# Influence of Post-Flowering Climate Conditions on Anthocyanin Profile of Strawberry Cultivars Grown from North to South Europe

Erika Krüger [1],*, Frank Will [2], Keshav Kumar [2], Karolina Celejewska [3], Philippe Chartier [4], Agnieszka Masny [3], Daniela Mott [5], Aurélie Petit [4], Gianluca Savini [5] and Anita Sønsteby [6]

1 Institute of Pomology, Hochschule Geisenheim University, 65366 Geisenheim, Germany
2 Institute of Beverage Research, Hochschule Geisenheim University, 65366 Geisenheim, Germany; Frank.Will@hs-gm.de (F.W.); Keshav.Kumar@hs-gm.de (K.K.)
3 The National Institute of Horticultural Research (INHORT), Konstytucji 3 Maja 1/3, 96-100 Skierniewice, Poland; karolina.celejewska@inhort.pl (K.C.); agnieszka.masny@inhort.pl (A.M.)
4 INVENIO, Maison Jeannette, 24140 Douville, France; p.chartier@invenio-fl.fr (P.C.); a.petit@invenio-fl.fr (A.P.)
5 Sant'Orsola Società Cooperativa Agricola, Via Lagorai, 127, 38057 Pergine Valsugana, Italy; Daniela.Mott@santorsola.com (D.M.); Gianluca.Savini@santorsola.com (G.S.)
6 NIBIO, Norwegian Institute of Bioeconomy Research, 1431 Ås, Norway; Anita.Sonsteby@Nibio.no
* Correspondence: Erika.Krueger@hs-gm.de

**Abstract:** The effect of cultivar and environmental variations and their interaction on anthocyanin components of strawberry were assessed for six cultivars grown in five locations from North to South of Europe in two different years. To evaluate the impact of latitude- and altitude-related factors, daily mean ($T_{mean}$), maximum ($T_{max}$) and minimum ($T_{min}$) temperature and global radiation accumulated for 3, 5, 10 and 15 days before fruit sampling, was analyzed. In general, fruits grown in the south were more enriched in total anthocyanin and pelargonidin-3-glucoside (pel-3-glc), the most abundant anthocyanin in strawberry. Principal component analysis (PCA) provided a separation of the growing locations within a cultivar due to latitudinal climatic differences, temporary weather changes before fruit collection and cultivation technique. PCA also depicted different patterns for anthocyanin distribution indicating a cultivar specific reaction on the environmental factors. The linear regression analysis showed that pel-3-glc was relatively less affected by these factors, while the minor anthocyanins cyanidin-3-glucoside, cyanidin-3-(6-O-malonyl)-glucoside, pelargonidin-3-rutinoside and pelargonidin-3-(6-O-malonoyl)-glucoside were sensitive to $T_{max}$. The global radiation strongly increased cya-3-mal-glc in 'Frida' and pel-3-rut in 'Frida' and 'Florence'. 'Candonga' accumulated less pel-3-glc and total anthocyanin with increased global radiation. The anthocyanin profiles of 'Gariguette' and 'Clery' were unaffected by environmental conditions.

**Keywords:** anthocyanins; *Fragaria × ananassa*; latitude; temperature; global radiation; cultivar × environmental interaction

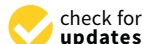

## 1. Introduction

Strawberry (*Fragaria × ananassa* Duch.) is the most important berry crop being cultivated from North to South of Europe. Beside its unique color, taste and aroma, strawberry fruits are enriched with several nutritious and bioactive compounds providing health benefits by reducing risk of diseases such as inflammation disorders and oxidative stress, obesity-related disorders and heart disease, and protection against various types of cancer [1–4]. Anthocyanins are a type of flavonoids that are commonly found in strawberries. The functional properties and the sensory qualities of the anthocyanins could easily be explained based on their chemical reactivity [5]. The antioxidative activity of anthocyanins could mainly be attributed to the presence of the flavylium cation moiety. Despite their low bioavailability [6], anthocyanins have been shown to exhibit a range of biological effects,

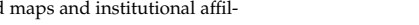

including antioxidant activity, photoprotection, anti-carcinogenesis, induction of apoptosis, and prevention of DNA damage [3,7]. Anthocyanins also serve as visual attractants for pollinators and seed dispensers and play a crucial role in plant protection against biotic and abiotic stress, and hence, in adaptability to environmental conditions at site [8].

Thus, the total anthocyanin contents in strawberry are both qualitatively and quantitatively known to be strongly influenced by the genotype (among others [9–14]) and likewise, by external factors such as high or low temperature and light (photoperiod, quantity and wavelength including UV-light). Recent review articles have highlighted the influence of temperature and light on the synthesis and accumulation of plant secondary metabolites including anthocyanins [15–17]. For example, some studies reported positive correlation between anthocyanin contents and temperature in strawberries grown in controlled environment [18–20], as well as in ambient conditions [14,21]. The studies describing the influence of incident light are mostly related to protected cultivation systems and includes shading [22], UV-B radiation [23–26] and blue and red LED-light [27,28].

The synthesis and accumulation of anthocyanins in strawberries are primarily known to be influenced by the genotype, however, little is known about latitudinal effects on the anthocyanin content of strawberry causing quantitative or qualitative changes in the content of these compounds [21,23]. The ambient conditions, including temperature and photoperiod, varies with latitude, and may also influence the different anthocyanin components. In future, due to the climate changes, the northern regions of Europe will become more suitable for berry production [25]. Moreover, it is desirable that quality traits of a cultivar, including bioactive compounds such as anthocyanins, remain invariant to the changes caused by environmental conditions during the cultivation season in different years at all growing locations. Cultivars with such stable fruit qualities could serve as useful parents in breeding programs with regard to climate warming.

The objective of the present work was to study the cultivar and environmental impact and their interaction on the anthocyanin content of six strawberry cultivars: 'Candonga', 'Clery', 'Frida', 'Florence', 'Gariguette' and 'Sonata' grown at five geographical distant locations throughout Europe from South-East Norway to South-West France, covering a distance of more than 15.5 degree of latitudes or more than 2000 km. These cultivars are mainly selected because of their diversity and popularity in Europe. The present study would help in identifying the appropriate cultivar with desirable anthocyanin traits that could be cultivated in a given environmental condition.

## 2. Materials and Methods

### 2.1. Experimental Sites, Plant Material and Cultivation

The experiments were conducted during two consecutive cropping seasons (2017 and 2018) at five locations from North to South Europe at NIBIO (Norwegian Institute of Bioeconomy Research, Bergen, Norway) (NO, 60° N), INHORT (PL, 51° N), HGU (DE, 49° N), Sant'Orsola (IT, 46° N) and Invenio (FR, 44° N), hereinafter named as Norway, Poland, Germany, Italy and France. As common for the different regions, experiments in Norway, Poland and Germany were carried out in open field, whereas in Italy and France they were performed in polytunnels that were open-sided after anthesis. Details of the respective latitude, altitude, yearly mean temperature, soil type, soil pH and cultivation type, as well as start of flowering, harvest season and day length at start of harvest are given in Table 1. Air temperature ($T_{mean}$, $T_{max}$ and $T_{min}$) and global radiation were recorded at each location. To describe the environmental conditions of the respective harvest season, weekly mean temperature and global radiation were calculated starting 2 weeks prior to start of harvest of the earliest cultivar 'Clery' (Figure 1).

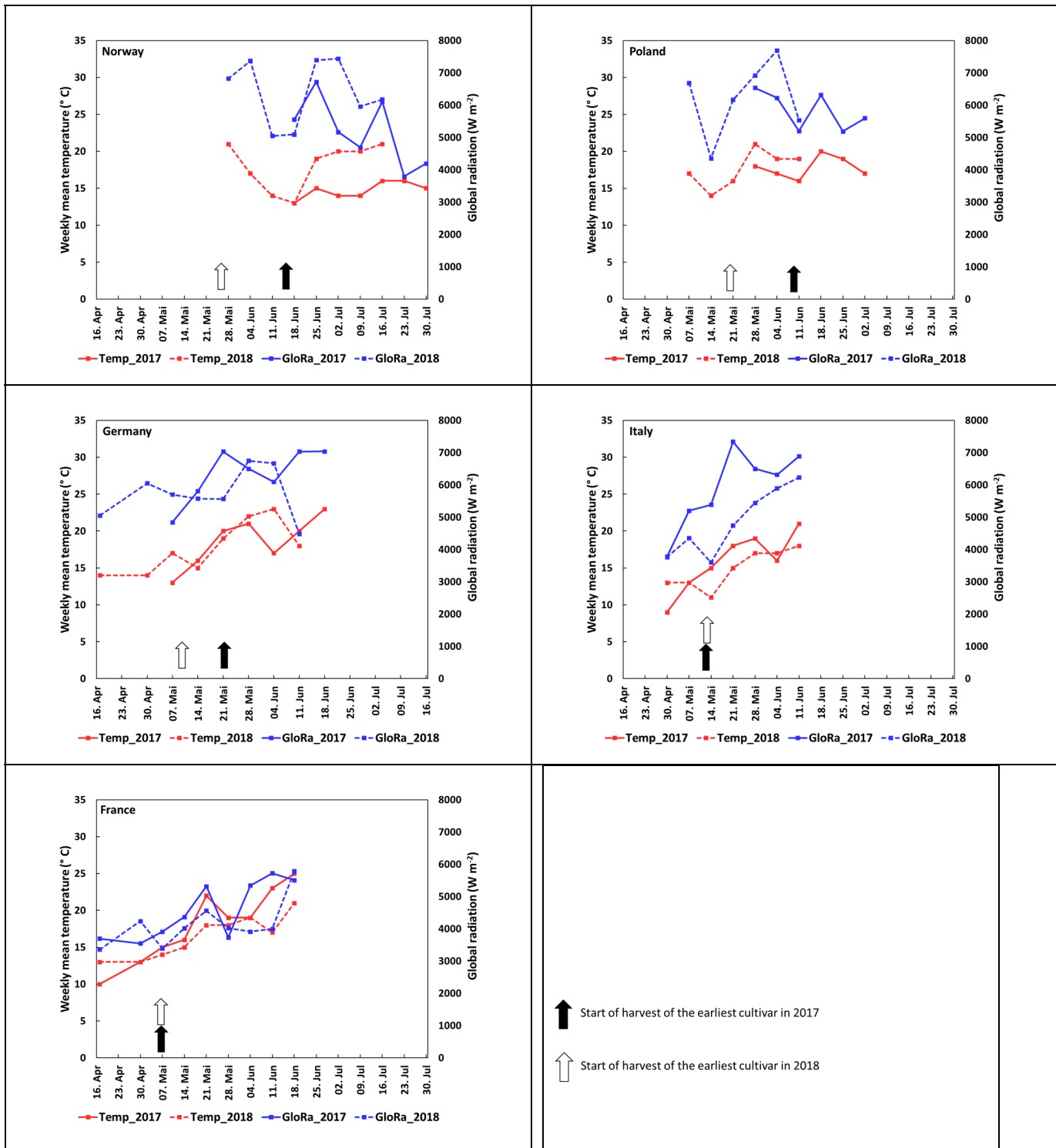

**Figure 1.** Weekly mean temperature and global radiation courses during the harvest season for the five experimental locations in 2017 and 2018, starting two weeks before the first picking of the earliest cultivar 'Clery'.

**Table 1.** Geographical location, soil and climatic conditions, cultivation type and harvest season for the five experimental locations in Europe.

| | NIBIO Norway | | INHORT Poland | | HGU Germany | | Sant'Orsola Italy | | INVENIO France | |
|---|---|---|---|---|---|---|---|---|---|---|
| Latitude | 60°40′ N | | 51°95′ N | | 49°59′ N | | 46°4′ N | | 44°85′ N | |
| Altitude (m a.s.l.) | 262 | | 252 | | 95 | | 925 | | 145 | |
| Yearly mean temperature (°C) [a] | 5.0 | | 7.9 | | 9.9 | | 11.3 | | 12.9 | |
| Soil type | Loam | | Pseudopodsol with light clay | | Sandy loam | | Soilless culture | | Soilless culture | |
| pH of the soil/substrate | 5.7–6.2 | | 6.5–7.0 | | 6.5–7.0 | | 5.5–6.0 | | 6.1 | |
| Cultivation type | open field | | open field | | open field | | tunnel | | tunnel | |
| | 2017 | 2018 | 2017 | 2018 | 2017 | 2018 | 2017 | 2018 | 2017 | 2018 |
| Yearly mean temperature (°C) [a] | 5.0 | 5.7 | 9.1 | 9.8 | 11.3 | 12.4 | 15.6 | 15.1 | 13.2 | 13.8 |
| Start of flowering [b] | 06.06. | 21.05. | 16.05 | 30.04. | 04.04. [d] | 16.04. | 04.04. | 11.04. | 04.04. | 07.04. |
| Start of harvest [b] | 07.07. | 17.06. | 13.06. | 22.05. | 29.05. | 14.05. | 19.05. | 17.05. | 09.05. | 09.05. |
| End of harvest [c] | 16.08. | 21.07. | 07.07. | 19.06. | 16.06. | 18.06. | 12.06. | 13.06. | 22.06. | 22.06. |
| Day length at start of harvest (h) [b] | 18:40 | 19:09 | 16:43 | 16:04 | 15:56 | 15:20 | 14:30 | 14:30 | 14.36 | 14.36 |

[a] For the period 1981–2010. [b] Earliest cultivar. [c] Latest cultivar. [d] Warm temperature in March enhanced start of flowering followed by extremely cold temperature delaying fruit development.

The six short-day strawberry genotypes used in this study were selected for their diverse genetic background and adaptability to different environments: 'Candonga'® (E), 'Clery' (IT), 'Florence' (UK), 'Frida' (NO), 'Gariguette' (FR) and 'Sonata' (NL). All cultivars were cultivated in Norway, Poland, Germany and France, while Italy only grew 'Clery', 'Frida', 'Gariguette' and 'Sonata'. Each season, all locations propagated their own plants from mother-plants purchased from the same nursery to avoid epigenetic variation within a cultivar due to different origin. Thus, each year, plug-plants were planted in week 32 in Norway, Poland and Germany for the harvest season the next year. At all sites, three randomized replicated plots, each with 20 plants per genotype, were trialed. In Italy and France, where strawberry are commonly grown in tunnels, cold-stored plants were set in peat bags, as usual, in week 9 in 2017 and 2018 for harvesting in the respective year. Furthermore, in Italy and France, the experimental design consisted of three plots, and of three replications per genotype with 22–24 plants per plot for the harvest in the same year. Plant protection, fertilization and irrigation in open-field sites, and fertigation of the plants in peat bags were performed according to local guidelines.

To evaluate the impact of latitude-related factors on the anthocyanin synthesis and accumulation, daily temperature ($T_{mean}$, $T_{max}$ and $T_{min}$) and global radiation were accumulated for 3, 5, 10 and 15 days (3d, 5d, 10d and 15d) before sampling of the berries to be analyzed for each cultivar.

### 2.2. Sample Preparation for Anthocyanin Analysis

Three independent biological replications were collected twice in an approximately weekly interval at mid-harvest at each location per cultivar (Figure 1 and Table 1). Thus, the samples (500 g) consisted mainly of secondary and tertiary fruits, being fully ripe and undamaged. Each biological replicate of freshly harvested strawberries was promptly prepared for analytical purposes by first cutting the calyx and then slicing the fruits into 2–4 pieces. Slices were frozen in a liquid nitrogen bath for at least 10 s. For anthocyanin analysis, the frozen slices were ground either in a lab mill (e.g., Retsch GM 200) or crushed under liquid nitrogen with a precooled mortar and pistil to a fine powder. The frozen powders were filled into 50 mL plastic tubes with screw caps, and stored at −20° C. Anthocyanin analysis was carried out at Geisenheim. Suitable shipping conditions were chosen to prevent thawing of the samples during transportation until solvent extraction. The frozen

powders (5.0 g) were weighed into a 50 mL plastic tube and extracted twice in an ultrasonic bath (30 min, 120 W, Bandelin Sonorex RK 106, VWR, Darmstadt, Germany) and intermediate centrifugation (4500 upm 15 min, Hettich Rotanta 460, Tuttlingen, Germany) with a total of $2 \times 10$ mL acidified methanol/water/formic acid (80/20/1 $v/v/v$). Sonication and centrifugation were performed under cool conditions. After final centrifugation, the pooled supernatants were made up to 25 mL in a volumetric flask; aliquots were 0.45 μm filtered prior to HPLC analysis.

### 2.3. HPLC Analysis of Anthocyanins

HPLC analysis of the methanolic extracts was performed on an Accela HPLC system coupled with a PDA detector (Thermo Fisher, Dreieich, Germany) using a $125 \times 2$ mm i.d., 3 μm ODS-3 column (Dr. Maisch, Ammerbuch, Germany) protected with a guard column of the same material. Injection volume was 4 μL, elution conditions were: 250 μL/min flow rate at 40 °C; solvent A was 5% formic acid (ULC/MS grade, Promochem, Wesel, Germany); solvent B, methanol (gradient grade, Roth, Karlsruhe, Germany); 1 min isocratic conditions with 10% B, linear gradient from 10% to 40% B in 12 min, followed by washing with 100% B and re-equilibrating the column. Quantitation was carried out using peak areas (500 nm trace for pelargonidins, 515 nm for cyanidins) from external calibration via the reference substances pelargonidin-3-glucoside and cyanidin-3-glucoside, respectively. Anthocyanin analysis was carried out in duplicate.

### 2.4. Electrospray Ionization (ESI)-MS Identification of Anthocyanins

For mass detection, the Accela HPLC system was coupled to a ThermoFinnigan LXQ mass spectrometer (Thermo Fisher, Dreieich, Germany) equipped with an ESI source and an ion trap mass analyzer. The whole system was controlled by Xcalibur software. For anthocyanins, the mass spectrometer was operated in the positive mode under the following conditions: source voltage 4.5 kV; capillary voltage 32 V; capillary temperature 275 °C; collision energy 30% ($MS^2$) and 33% ($MS^3$).

### 2.5. Software Used for Statistical Analysis

All statistical analysis was conducted on the MATLAB (2016b) platform. Significant differences were calculated using post hoc analysis with Tukey's honestly significant difference criteria on the ANOVA (analysis of variance) results. Principal component analysis (PCA) [29] was performed using the PLS-Toolbox (Eigenvector Research, Manson, WA, USA). Linear regression analysis was used to study the correlation between different cultivar-specific anthocyanin components and the mean, minimum, maximum temperatures, and global radiation summarized 3, 5, 10 and 15 days before start of harvest. Coefficient of determinations ($R^2$) $\geq 0.2$ at $p \leq 0.05$ are presented.

### 2.6. Data Arrangement for PCA

The data for 'Clery', 'Frida', 'Gariguette' and 'Sonata' grown at the locations in France, Italy, Germany, Poland and Norway were arranged in a matrix of dimension $60 \times 22$; 60 represents the number of samples (five locations, two cropping seasons, two picking dates and three biological replicates) and 22, the number of variables (cya-3-glc, pel-3-glc, pel-3-rut, cya-3-mal-glc, pel-3-mal-glc, total anthocyanin as well as global radiation, mean, maximum and minimum temperatures summarized 3, 5, 10 and 15 days prior to harvest). The data for 'Candonga' and 'Florence' were arranged in matrices of dimension $48 \times 22$, where 48 is the number of samples (four location × two seasons × two picking dates × three biological replicates) and 22 is the same variables as specified above. The PCA analysis carried out on all the cultivars together clearly indicated (given in the Supplementary Figure S1) that the anthocyanin profiles of the strawberries are mainly influenced by the location. Thus it was important that each of the six cultivars were analyzed separately.

To ensure that each component had equal variance and comparable impact on the PCA modeling, the specific data for each cultivar were auto-scaled prior to PCA. Autos-

scaling [29] is a common pre-processing method that uses mean-centering followed by dividing each variable by the corresponding standard deviation. Each variable upon auto-scaling has a mean of zero and unit variance.

## 3. Results and Discussion

### 3.1. Harvest Season and Environmental Characterization of the Growing Locations

This study investigated the adaption of six strawberry genotypes to different environments related to anthocyanin accumulation in the fruits. As expected, the harvest period varied along the North-South axis (Table 1), due to the longitudinal difference between the growing sites (>15.5° of latitude). In addition, yearly differences in temperature and global radiation affected the harvest period.

For instance, the harvest period was much earlier in 2018 compared to 2017 in Norway, Poland and Germany, while there were no variations in Italy and France. To highlight the environmental differences between the five growing sites, ambient weekly mean temperature and global radiation during the harvest seasons are shown in Figure 1, starting two weeks prior to start of harvest of the earliest cultivar 'Clery'. Higher year-on-year variations were observed in Norway. Here, weekly mean temperature was, on average, 3 °C lower, and global radiation 1200 W m$^{-2}$ less in 2017 compared to 2018 (18 °C and 6412 W m$^{-2}$). In this year, weekly mean temperature and global radiation in Norway were similar to those in both years in Poland and Germany, and for temperature in France, only. Due to its high altitude (Table 1), Italy exhibited lower weekly mean temperature in both years, thus being similar to Norway in 2017. In Italy, large yearly variations occurred for global radiation in 2017 (5917 W m$^{-2}$) with similar values as for Norway, Poland and Germany, while in 2018, Italy had the same low values as for both years in France. The low values for global radiation at the southern locations are probably due to the earlier cropping season and shorter photoperiod compared to Norway, as the other extreme.

The temperature and global radiation may vary within the harvest season, therefore, the sum of temperature and global radiation 3, 5, 10 and 15 days before the two fruit samplings were calculated for each cultivar and location, and used for principal component and regression analyses. The complete list is provided in the Supplementary Materials (Table S1).

### 3.2. Effect of Genotype on Total and Individual Anthocyanins

Cyanidin-3-glucoside (cya-3-glc, m/z [M$^+$] 449, MS$^2$ 287), pelargonidin-3-glucoside (pel-3-glc, m/z [M$^+$] 433, MS$^2$ 271), pelargonidin-3-rutinoside (pel-3-rut, m/z [M$^+$] 579, MS$^2$ 433, 271), cyanidin-3-(6-O-malonyl)-glucoside (cya-3-mal-glc, m/z [M$^+$] 535, MS$^2$ 287), and pelargonidin-3-(6-O-malonoyl)-glucoside (pel-3-mal-glc, m/z [M$^+$] 519, MS$^2$ 271) were assigned by their mass spectra as the major strawberry anthocyanins (Tables 2–7). Over all, the pel-3-glc showed the highest concentrations (Table 4).

**Table 2.** Effect of growing location and year on total anthocyanin content as the mean of two picking dates in fruits of six strawberry cultivars.

| | | Total Anthocyanins HPLC (mg kg$^{-1}$ Fresh Weight) | | | | | | |
|---|---|---|---|---|---|---|---|---|
| | | **Cultivar** | | | | | | |
| **Location** | **Year** | **Can** | **Cle** | **Flo** | **Fri** | **Gar** | **Son** | *Yearly Mean per Location* |
| **Norway** | **2017** | 212.6 ± 25.6 | 284.6 ± 30.5 | 374.5 ± 30.4 | 453.4 ± 42.4 | 263.1 ± 58.2 | 191.6 ± 16.2 | *296.6 B* |
| | **2018** | 133.8 ± 33.8 | 195.2 ± 48.7 | 405.5 ± 43.8 | 451.6 ± 68.8 | 198.9 ± 25.8 | 145.6 ± 30.4 | *255.1 B* |
| | *mean* | *173.2 ab* | *239.9 c* | *390.0 d* | *452.5 e* | *231.0 bc* | *168.6 a* | |
| **Poland** | **2017** | 245.9 ± 53.0 | 310.1 ± 34.3 | 359.1 ± 89.4 | 438.0 ± 58.8 | 235.0 ± 34.9 | 210.2 ± 27.2 | *299.7 B* |
| | **2018** | 225.0 ± 40.8 | 249.6 ± 22.4 | 448.5 ± 68.5 | 397.9 ± 74.6 | 217.6 ± 40.1 | 222.6 ± 35.4 | *293.5 B* |
| | *mean* | *235.5 a* | *279.9 a* | *403.8 b* | *417.9 b* | *226.3 a* | *216.4 a* | |

**Table 2.** *Cont.*

| | | Total Anthocyanins HPLC (mg kg$^{-1}$ Fresh Weight) | | | | | | |
|---|---|---|---|---|---|---|---|---|
| | | Cultivar | | | | | | |
| Location | Year | Can | Cle | Flo | Fri | Gar | Son | *Yearly Mean per Location* |
| **Germany** | **2017** | 196.5 ± 22.3 | 285.5 ± 46.5 | 437.3 ± 57.7 | 429.2 ± 21.1 | 225.5 ± 44.4 | 199.4 ± 44.2 | *295.6 B* |
| | **2018** | 223.6 ± 31.3 | 307.7 ± 21.1 | 507.9 ± 26.5 | 385.9 ± 27.9 | 273.5 ± 19.9 | 217.7 ± 51.1 | *319.4 B* |
| | *mean* | *211.0 a* | *296.6 b* | *472.6 d* | *407.6 c* | *249.5 ab* | *208.6 a* | |
| **Italy** | **2017** | - | 363.0 ± 27.2 | - | 432.4 ± 52.7 | 326.4 ± 19.4 | 209.2 ± 17.7 | *332.8 B* |
| | **2018** | - | 276.6 ± 44.9 | - | 302.2 ± 56.2 | 166.9 ± 22.3 | 124.7 ± 34.4 | *217.6 A* |
| | *mean* | - | *319.8 bc* | - | *367.4 c* | *246.6 b* | *167.0 a* | |
| **France** | **2017** | 315.5 ± 23.2 | 218.9 ± 63.8 | 455.2 ± 81.3 | 446.7 ± 44.3 | 261.5 ± 39.3 | 212.0 ± 48.3 | *318.3 B* |
| | **2018** | 306.9 ± 45.2 | 365.6 ± 61.3 | 420.9 ± 66.3 | 342.8 ± 64.3 | 221.2 ± 34.4 | 198.6 ± 49.2 | *309.2 B* |
| | *mean* | *311.2 b* | *292.3 b* | *438.0 c* | *394.4 c* | *241.3 ab* | *205.3 a* | |
| *Cultivar mean over all locations* | | *232.5 b* | *285.7 c* | *426.1 d* | *408.0 d* | *239.0 b* | *193.1 a* | |
| *Significance* | | *Cultivar* | *Location* | *Year* | *Cultivar x location* | *Cultivar x year* | *Location x year* | |
| | | *** | *ns* | * | *ns* | *ns* | * | |

Data are expressed as means ± SD (standard deviation) of two sampling dates per year. Before performing the statistical analysis, the homogeneity of the data were ensured using Bartlett's test. Mean values (*n* = 3) of different cultivars grown at a particular location followed by lower-case letters represent significant differences ($p \leq 0.05$) between cultivars. Mean values of all the cultivars grown in a particular location followed by different upper-case letters represent significant difference between the two years 2017 and 2018 ($p \leq 0.05$). Mean values of all cultivars grown at all the locations followed by different lower-case letters represent significant differences ($p \leq 0.05$). * = 0.05; *** = 0.001; ns = not significant. Can = 'Candonga'; Cle = 'Clery'; Flo = 'Florence'; Fri = 'Frida'; Gar = 'Gariguette'; Son = 'Sonata'.

**Table 3.** Effect of growing location and year on cyanidin 3-glucoside content as the mean of two picking dates in fruits of six strawberry cultivars.

| | | Cyanidin 3-Glucoside (mg kg$^{-1}$ Fresh Weight) | | | | | | |
|---|---|---|---|---|---|---|---|---|
| | | Cultivar | | | | | | |
| Location | Year | Can | Cle | Flo | Fri | Gar | Son | *Yearly Mean per Location* |
| **Norway** | **2017** | 9.6 ± 2.6 | 3.6 ± 1.5 | 9.4 ± 2.8 | 14.1 ± 2.4 | 3.2 ± 0.7 | 5.7 ± 1.7 | *7.6 A* |
| | **2018** | 12.7 ± 3 | 7.4 ± 5.2 | 14.3 ± 2.2 | 13.2 ± 1.1 | 4.4 ± 1.2 | 6.0 ± 2.3 | *9.6 A* |
| | *mean* | *11.2 b* | *5.5 a* | *11.8 b* | *13.7 b* | *3.8 a* | *5.8 a* | |
| **Poland** | **2017** | 9.7 ± 1.7 | 3.2 ± 0.6 | 12.7 ± 4.5 | 15.0 ± 3.3 | 3.3 ± 0.5 | 6.3 ± 2.6 | *8.4 A* |
| | **2018** | 29.2 ± 2.6 | 8.8 ± 1.9 | 47.8 ± 6.7 | 27.8 ± 4.1 | 9.2 ± 2.1 | 13.4 ± 5.9 | *22.7 B* |
| | *mean* | *19.4 bc* | *6.0 a* | *30.3 c* | *21.4 bc* | *6.2 a* | *9.9 ab* | |
| **Germany** | **2017** | 10.7 ± 2.4 | 2.5 ± 0.6 | 17.4 ± 3.6 | 12.8 ± 1.2 | 2.5 ± 1.3 | 4.4 ± 1.6 | *8.4 A* |
| | **2018** | 12.9 ± 4.8 | 5.0 ± 4.8 | 26.4 ± 4.4 | 16.9 ± 2.0 | 4.0 ± 0.5 | 6.0 ± 2.4 | *11.9 A* |
| | *mean* | *11.8 b* | *3.7 a* | *21.9 c* | *14.8 b* | *3.3 a* | *5.2 a* | |
| **Italy** | **2017** | - | 1.6 ± 0.3 | - | 11.9 ± 4.6 | 2.5 ± 0.5 | 2.8 ± 0.5 | *4.7 B* |
| | **2018** | - | 1.6 ± 0.7 | - | 2.4 ± 1.5 | 1.5 ± 0.7 | 2.7 ± 1.8 | *2.0 A* |
| | *mean* | - | *1.6 a* | - | *7.0 b* | *2.0 a* | *2.7 a* | |
| **France** | **2017** | 11.3 ± 1.5 | 1.3 ± 0.1 | 19.0 ± 14.5 | 12.5 ± 1.7 | 1.7 ± 0.3 | 2.6 ± 0.7 | *8.0 A* |
| | **2018** | 10.7 ± 3.0 | 1.8 ± 0.4 | 16.9 ± 5.8 | 9.0 ± 2.1 | 2.7 ± 0.6 | 3.1 ± 1.3 | *7.4 A* |
| | *mean* | *11.0 b* | *1.6 a* | *18.0 c* | *10.7 b* | *2.2 a* | *2.8 a* | |
| *Cultivar mean over all locations* | | *13.3 b* | *3.7 a* | *20.5 c* | *13.5 b* | *3.5 a* | *5.3 a* | |

**Table 3.** *Cont.*

| | | Cyanidin 3-Glucoside (mg kg$^{-1}$ Fresh Weight) | | | | | | |
|---|---|---|---|---|---|---|---|---|
| | | **Cultivar** | | | | | | |
| **Location** | **Year** | **Can** | **Cle** | **Flo** | **Fri** | **Gar** | **Son** | *Yearly Mean per Location* |
| *Significance* | | *Cultivar* *** | *Location r* *** | *Year* *** | *Cultivar x location* *ns* | *Cultivar x year* *ns* | *Location x year* *** | |

Data are expressed as means ± SD (standard deviation) of two sampling dates per year. Before performing the statistical analysis, the homogeneity of the data were ensured using Bartlett's test. Mean values (*n* =3) of different cultivars grown at a particular location followed by lower-case letters represent significant differences ($p \leq 0.05$) between cultivars. Mean values of all the cultivars grown in a particular location followed by different upper-case letters represent significant difference between the two years 2017 and 2018 ($p \leq 0.05$). Mean values of all cultivars grown at all the locations followed by different lower-case letters represent significant differences ($p \leq 0.05$). *** = 0.001; ns = not significant. Can = 'Candonga'; Cle = 'Clery'; Flo = 'Florence'; Fri = 'Frida'; Gar = 'Gariguette'; Son = 'Sonata'.

**Table 4.** Effect of growing location and year on pelargonidin-3-glucoside content as the mean of two picking dates in fruits of six strawberry cultivars.

| - | | Pelargonidin-3-glucoside (mg kg$^{-1}$ Fresh Weight) | | | | | | |
|---|---|---|---|---|---|---|---|---|
| | | **Cultivar** | | | | | | |
| **Location** | **Year** | **Can** | **Cle** | **Flo** | **Fri** | **Gar** | **Son** | *Yearly Mean per Location* |
| **Norway** | **2017** | 184.3 ± 23.9 | 228.4 ± 23.1 | 330.3 ± 26.7 | 328.1 ± 36.7 | 178.97 ± 39.8 | 145.4 ± 13.9 | *232.6 A* |
| | **2018** | 108.8 ± 27.7 | 146.5 ± 41.2 | 346.0 ± 35.6 | 340.1 ± 39.2 | 133.67 ± 16.2 | 95.4 ± 48.9 | *195.1 A* |
| | *mean* | *146.6 ab* | *187.4 b* | *338.1 c* | *334.1 c* | *156.3 ab* | *120.4 a* | |
| **Poland** | **2017** | 209.9 ± 48.0 | 246.2 ± 24.0 | 306.0 ± 85 | 306.5 ± 37.4 | 160.4 ± 24.1 | 154.3 ± 17.2 | *230.6 A* |
| | **2018** | 162.2 ± 15.2 | 177.9 ± 18.5 | 333.4 ± 54.6 | 253.2 ± 50.8 | 131.6 ± 25.5 | 146.7 ± 21.3 | *200.8 A* |
| | *mean* | *186.0 ab* | *212.1 b* | *319.7 c* | *279.9 c* | *146.0 a* | *150.5 a* | |
| **Germany** | **2017** | 164.5 ± 17.6 | 216.4 ± 35.3 | 382.3 ± 49.8 | 292.4 ± 12.7 | 149.8 ± 24.3 | 140.7 ± 26.6 | *224.3 A* |
| | **2018** | 189.5 ± 21.6 | 233.2 ± 15.7 | 420.8 ± 16.8 | 268.1 ± 18.8 | 183.8 ± 9.0 | 158.4 ± 37.6 | *242.1 A* |
| | *mean* | *177.0 a* | *224.8 b* | *401.5 d* | *280.3 c* | *166.8 a* | *149.5 a* | |
| **Italy** | **2017** | - | 288.5 ± 22.4 | - | 306.2 ± 41.7 | 222.2 ± 19.9 | 160.1 ± 15.8 | *244.3 B* |
| | **2018** | - | 200.8 ± 33.4 | - | 221.0 ± 39.5 | 102.1 ± 17.5 | 91.4 ± 24.0 | *153.8 A* |
| | *mean* | - | *244.7 b* | - | *263.8 b* | *162.1 a* | *125.8 a* | |
| **France** | **2017** | 279.5 ± 21.6 | 171.1 ± 46.4 | 395.0 ± 50.6 | 321.8 ± 33.1 | 176.0 ± 27.8 | 159.9 ± 36.1 | *250.5 B* |
| | **2018** | 271.0 ± 39.5 | 274.0 ± 53.1 | 345.7 ± 55.5 | 225.2 ± 42.7 | 125.5 ± 31.6 | 140.7 ± 33.6 | *230.4 A* |
| | *mean* | *275.2 b* | *222.6 b* | *370.3 c* | *273.5 b* | *150.8 a* | *150.3 a* | |
| *Cultivar mean over all locations* | | *196.2 b* | *218.3 b* | *357.4 d* | *286.3 c* | *156.4 a* | *139.3 a* | |
| *Significance* | | *Cultivar* *** | *Location* *ns* | *Year* *** | *Cultivar x location* *ns* | *Cultivar x year* *ns* | *Location x Year* * | |

Data are expressed as means ± SD (standard deviation) of two sampling dates per year. Before performing the statistical analysis, the homogeneity of the data were ensured using Bartlett's test. Mean values (*n* = 3) of different cultivars grown at a particular location followed by lower-case letters represent significant differences ($p \leq 0.05$) between cultivars. Mean values of all the cultivars grown in a particular location followed by different upper-case letters represent significant difference between the two years 2017 and 2018 ($p \leq 0.05$). Mean values of all cultivars grown at all the locations followed by different lower-case letters represent significant differences ($p \leq 0.05$). * = 0.05; *** = 0.001; ns = not significant. Can = 'Candonga'; Cle = 'Clery'; Flo = 'Florence'; Fri = 'Frida'; Gar = 'Gariguette'; Son = 'Sonata'.

**Table 5.** Effect of growing location and year on pelargonidin-3-rutinoside content as the mean of two picking dates in fruits of six strawberry cultivars.

| | | Pelargonidin-3-rutinoside (mg kg$^{-1}$ Fresh Weight) | | | | | | |
|---|---|---|---|---|---|---|---|---|
| | | Cultivar | | | | | | |
| Location | Year | Can | Cle | Flo | Fri | Gar | Son | *Yearly Mean per Location* |
| **Norway** | **2017** | 18.2 ± 2.5 | 5.3 ± 1.1 | 1.4 ± 0.2 | 27.0 ± 3.7 | 0.9 ± 0.5 | 7.3 ± 1.1 | *10.0 A* |
| | **2018** | 11.9 ± 3.4 | 2.5 ± 1.5 | 1.4 ± 0.2 | 23.3 ± 11.9 | 0.5 ± 0.1 | 19.2 ± 3.2 | *9.8 A* |
| | *mean* | *15.1 cd* | *3.9 abc* | *1.4 ab* | *25.2 d* | *0.7 a* | *13.2 bcd* | |
| **Poland** | **2017** | 26.2 ± 4.4 | 7.1 ± 0.4 | 2.1 ± 1.7 | 25.2 ± 3.2 | 0.9 ± 0.3 | 5.9 ± 0.9 | *11.2 A* |
| | **2018** | 28.3 ± 5.7 | 9.1 ± 1.1 | 3.7 ± 0.3 | 25.9 ± 4.8 | 3.0 ± 0.2 | 8.5 ± 0.6 | *13.1 A* |
| | *mean* | *27.3 c* | *8.1 b* | *2.9 a* | *25.5 c* | *2.0 a* | *7.2 b* | |
| **Germany** | **2017** | 21.0 ± 3.6 | 6.7 ± 0.7 | 1.4 ± 0.3 | 25.4 ± 1.8 | 0.9 ± 0.5 | 6.4 ± 1.9 | *10.3 A* |
| | **2018** | 21.2 ± 9.6 | 9.6 ± 6.8 | 6.2 ± 1.5 | 20.9 ± 3.6 | 2.3 ± 0.7 | 6.8 ± 2.5 | *11.2 A* |
| | *mean* | *21.1 c* | *8.2 b* | *3.8 ab* | *23.1 c* | *1.6 a* | *6.6 b* | |
| **Italy** | **2017** | - | 8.1 ± 0.9 | - | 24.0 ± 3.8 | 2.0 ± 0.2 | 5.5 ± 0.5 | *9.9 B* |
| | **2018** | - | 7.1 ± 1.2 | - | 7.1 ± 2.5 | 1.3 ± 0.1 | 2.6 ± 1.2 | *4.5 A* |
| | *mean* | - | *7.6 b* | - | *15.6 c* | *1.6 a* | *4.1 ab* | |
| **France** | **2017** | 23.6 ± 3.2 | 4.5 ± 1.6 | 2.0 ± 0.1 | 20.4 ± 3.7 | 1.2 ± 0.4 | 4.8 ± 1.8 | *9.4 A* |
| | **2018** | 24.3 ± 3.1 | 7.2 ± 1.3 | 1.8 ± 0.4 | 14.1 ± 3.8 | 1.8 ± 0.9 | 3.6 ± 1.5 | *8.8 A* |
| | *mean* | *23.9 d* | *5.8 b* | *1.9 a* | *17.2 c* | *1.5 a* | *4.2 ab* | |
| *Cultivar mean over all locations* | | *21.8 c* | *6.7 b* | *2.5 a* | *21.3 c* | *1.5 a* | *7.0 b* | |
| *Significance* | | *Cultivar* ***  | *Location* ns | *Year* ns | *Cultivar x location* ** | *Cultivar x year* * | *Location x year* ns | |

Data are expressed as means ± SD (standard deviation) of two sampling dates per year. Before performing the statistical analysis, the homogeneity of the data were ensured using Bartlett's test. Mean values (*n* = 3) of different cultivars grown at a particular location followed by lower-case letters represent significant differences ($p \leq 0.05$) between cultivars. Mean values of all the cultivars grown in a particular location followed by different upper-case letters represent significant difference between the two years 2017 and 2018 ($p \leq 0.05$). Mean values of all cultivars grown at all the locations followed by different lower-case letters represent significant differences ($p \leq 0.05$). * = 0.05; ** = 0.01; *** = 0.001; ns = not significant. Can = 'Candonga'; Cle = 'Clery'; Flo = 'Florence'; Fri = 'Frida'; Gar = 'Gariguette'; Son = 'Sonata'.

**Table 6.** Effect of growing location and year on cyanidin-3-(6-O-malonyl)-glucoside content as the mean of two picking dates in fruits of six strawberry cultivars.

| | | Cyanidin-3-(6-O-malonyl)-glucoside (mg kg$^{-1}$ Fresh Weight) | | | | | | |
|---|---|---|---|---|---|---|---|---|
| | | Cultivar | | | | | | |
| Location | Year | Can | Cle | Flo | Fri | Gar | Son | *Yearly Mean per Location* |
| **Norway** | **2017** | 0.1 ± 0.2 | 0.7 ± 0.2 | 0.8 ± 0.2 | 3.3 ± 1.5 | 1.4 ± 0.6 | 1.2 ± 0.5 | *1.3 A* |
| | **2018** | 0.2 ± 0.1 | 1.5 ± 0.9 | 2.6 ± 0.5 | 2.8 ± 0.4 | 2.3 ± 0.5 | 0.8 ± 0.7 | *1.7 A* |
| | *mean* | *0.2 a* | *1.1 b* | *1.7 b* | *3.0 c* | *1.8 b* | *1.0 ab* | |
| **Poland** | **2017** | 0.0 ± 0.0 | 0.7 ± 0.2 | 1.6 ± 0.8 | 3.5 ± 1.7 | 1.2 ± 0.5 | 1.4 ± 0.9 | *1.4 A* |
| | **2018** | 2.5 ± 0.0 | 4.4 ± 0.4 | 11.0 ± 1.6 | 8.6 ± 1.2 | 5.6 ± 1.2 | 5.6 ± 1.0 | *6.3 B* |
| | *mean* | *1.2 a* | *2.5 a* | *6.3 b* | *6.1 b* | *3.4 ab* | *3.5 ab* | |
| **Germany** | **2017** | 0.0 ± 0.0 | 0.7 ± 0.2 | 2.3 ± 0.6 | 3.8 ± 0.3 | 1.2 ± 1.1 | 1.3 ± 0.7 | *1.6 A* |
| | **2018** | 0.0 ± 0.0 | 0.8 ± 0.4 | 3.5 ± 0.7 | 4.1 ± 0.9 | 1.7 ± 0.2 | 1.6 ± 0.8 | *2.0 A* |
| | *mean* | *0.0 a* | *0.8 b* | *2.9 c* | *4.0 d* | *1.5 b* | *1.5 b* | |
| **Italy** | **2017** | - | 0.7 ± 0.1 | - | 2.6 ± 1.4 | 0.9 ± 0.4 | 0.8 ± 0.1 | *1.2 B* |
| | **2018** | - | 0.7 ± 0.2 | - | 0.7 ± 0.4 | 0.7 ± 0.3 | 0.7 ± 0.4 | *0.7 A* |
| | *mean* | - | *0.7 a* | - | *1.7 b* | *0.8 a* | *0.8 a* | |

**Table 6.** *Cont.*

| | | Cyanidin-3-(6-O-malonyl)-glucoside (mg kg$^{-1}$ Fresh Weight) | | | | | | |
|---|---|---|---|---|---|---|---|---|
| | | **Cultivar** | | | | | | |
| **Location** | **Year** | **Can** | **Cle** | **Flo** | **Fri** | **Gar** | **Son** | *Yearly Mean per Location* |
| **France** | **2017** | 0.1 ± 0.2 | 0.2 ± 0.1 | 1.6 ± 0.9 | 3.1 ± 0.3 | 0.5 ± 0.2 | 0.7 ± 0.2 | *1.0 A* |
| | **2018** | 0.3 ± 0.1 | 0.6 ± 0.2 | 3.3 ± 1.4 | 3.1 ± 0.6 | 1.7 ± 0.4 | 1.0 ± 0.4 | *1.7 B* |
| | *mean* | *0.2 a* | *0.4 ab* | *2.5 c* | *3.1 c* | *1.1 b* | *0.8 ab* | |
| *Cultivar mean over all locations* | | *0.4 a* | *1.1 ab* | *3.4 c* | *3.6 c* | *1.7 b* | *1.5 b* | |
| *Significance* | | *Cultivar* | *Location* | *Year* | *Cultivar x location* | *Cultivar x year* | *Location x year* | |
| | | *** | *** | *** | ns | * | *** | |

Data are expressed as means ± SD (standard deviation) of two sampling dates per year. Before performing the statistical analysis, the homogeneity of the data were insured using Bartlett's test. Mean values (*n* =3) of different cultivars grown at a particular location followed by lower-case letters represent significant differences ($p \leq 0.05$) between cultivars. Mean values of all the cultivars grown in a particular location followed by different upper-case letters represent significant difference between the two years 2017 and 2018 ($p \leq 0.05$). Mean values of all cultivars grown at all the locations followed by different lower-case letters represent significant differences ($p \leq 0.05$). * = 0.05; *** = 0.001; ns = not significant. Can = 'Candonga'; Cle = 'Clery'; Flo = 'Florence'; Fri = 'Frida'; Gar = 'Gariguette'; Son = 'Sonata'.

**Table 7.** Effect of growing location and year on pelargonidin-3-(6-O-malonyl)-glucoside content as the mean of two picking dates in fruits of six strawberry cultivars.

| | | Pelargonidin-3-(6-O-malonyl)-glucoside (mg kg$^{-1}$ Fresh Weight) | | | | | | |
|---|---|---|---|---|---|---|---|---|
| | | **Cultivar** | | | | | | |
| **location** | **Year** | **Can** | **Cle** | **Flo** | **Fri** | **Gar** | **Son** | *Yearly Mean per Location* |
| **Norway** | **2017** | 0.4 ± 0.6 | 46.7 ± 7.7 | 32.6 ± 5.9 | 80.9 ± 10.3 | 78.7 ± 19.1 | 32.0 ± 5.3 | *45.2 A* |
| | **2018** | 0.2 ± 0.1 | 37.3 ± 8.6 | 41.3 ± 7.9 | 72.2 ± 24.5 | 58.0 ± 9.3 | 24.2 ± 5.5 | *38.9 A* |
| | *mean* | *0.3 a* | *42.0 b* | *36.9 b* | *76.5 c* | *68.3 c* | *28.1 b* | |
| **Poland** | **2017** | 0.2 ± 0.4 | 52.9 ± 10.4 | 36.6 ± 8.0 | 87.8 ± 17.9 | 69.2 ± 11.8 | 42.4 ± 7.5 | *48.2 A* |
| | **2018** | 2.8 ± 0.3 | 49.4 ± 3 | 52.5 ± 9.9 | 82.4 ± 16.1 | 68.1 ± 12.7 | 48.4 ± 8.2 | *50.6 A* |
| | *mean* | *1.5 a* | *51.2 b* | *44.5 b* | *85.1 d* | *68.6 c* | *45.4 b* | |
| **Germany** | **2017** | 0.4 ± 0.3 | 59.2 ± 12.8 | 33.9 ± 16.7 | 94.8 ± 10.5 | 71.1 ± 18.8 | 46.7 ± 14.4 | *51.0 A* |
| | **2018** | 0.0 ± 0.0 | 59.2 ± 6.4 | 51.1 ± 8.5 | 75.9 ± 12.8 | 81.6 ± 10.6 | 44.9 ± 11.8 | *52.1 A* |
| | *mean* | *0.2 a* | *59.2 c* | *42.5 b* | *85.4 d* | *76.4 d* | *45.8 b* | |
| **Italy** | **2017** | - | 64.1 ± 11.0 | - | 87.8 ± 13.0 | 98.8 ± 7.7 | 40.0 ± 5.9 | *72.6 B* |
| | **2018** | - | 66.3 ± 13 | - | 71.5 ± 13.3 | 61.3 ± 8.8 | 27.3 ± 7.4 | *56.6 A* |
| | *mean* | - | *65.2 b* | - | *79.6 b* | *80.1 b* | *33.6 a* | |
| **France** | **2017** | 1.1 ± 1.5 | 41.8 ± 16.0 | 37.7 ± 29.4 | 88.8 ± 14.3 | 82.1 ± 13.2 | 44.2 ± 11.8 | *49.3 A* |
| | **2018** | 0.7 ± 0.3 | 82.0 ± 14.8 | 53.1 ± 10.1 | 90.9 ± 17.8 | 89.4 ± 24.4 | 50.3 ± 15.6 | *61.1 A* |
| | *mean* | 0.9 a | 61.9 b | 45.4 b | 89.9 c | 85.8 c | 47.2 b | |
| *Cultivar mean over all locations* | | *0.7 a* | *55.9 c* | *42.3 b* | *83.3 d* | *75.8 d* | *40.2 b* | |
| *significance* | | *Cultivar* | *Location* | *Year* | *Cultivar x location* | *Cultivar x year* | *Location x Year* | |
| | | *** | *** | ns | ns | ns | * | |

Data are expressed as means ± SD (standard deviation) of two sampling dates per year. Before performing the statistical analysis, the homogeneity of the data were ensured using Bartlett's test. Mean values (*n* =3) of different cultivars grown at a particular location followed by lower-case letters represent significant differences ($p \leq 0.05$) between cultivars. Mean values of all the cultivars grown in a particular location followed by different upper-case letters represent significant difference between the two years 2017 and 2018 ($p \leq 0.05$). Mean values of all cultivars grown at all the locations followed by different lower-case letters represent significant differences ($p \leq 0.05$). * = 0.05; *** = 0.001; ns = not significant. Can = 'Candonga'; Cle = 'Clery'; Flo = 'Florence'; Fri = 'Frida'; Gar = 'Gariguette'; Son = 'Sonata'.

The genotype influenced significantly ($p \leq 0.05$) the abundance of total and individual anthocyanins in the fruits (Tables 2–7). When considering the cultivar means for all locations, 'Florence' and 'Frida' showed the highest total anthocyanin content (426.1 and 408.0 mg kg$^{-1}$ fresh weight (FW), respectively) whereas 'Sonata' had the lowest (193.2 mg kg$^{-1}$ FW) (Table 2). As expected, total anthocyanin content was strongly related to the predominant anthocyanin pel-3-glc (Table 4), resulting in the highest content for 'Florence' (357.4 mg kg$^{-1}$ FW) and, however, being significantly different from 'Florence', for 'Frida' (286.3 mg kg$^{-1}$ FW) and the lowest for 'Sonata' (139.3 mg kg$^{-1}$ FW) and also for 'Gariguette' (156.4 mg kg$^{-1}$ FW). Thereby, pel-3-glc contribution to the total anthocyanin content in the cultivars was in the range of 66–84%. 'Florence' (20.5 mg kg$^{-1}$ FW) also had, on average, a higher content of cya-3-glc compared to the other cultivars (Table 3). Pel-3-rut was highest in 'Candonga' and 'Frida' (21.8 and 21.3 mg kg$^{-1}$ FW) (Table 5), whereas cya-3-mal-glc was enriched in 'Florence' and 'Frida' (20.5 and 13.5 mg kg$^{-1}$ FW) (Table 6). The level of pel-3-mal-glc, the second abundant anthocyanin in strawberry, was again highest in 'Frida' (83.3 mg kg$^{-1}$ FW) and in 'Gariguette' (75.8 mg kg$^{-1}$ FW) (Table 7). The contribution of pel-3-mal-glc varied widely in a range of 9.9–31.7% of the total anthocyanin. In contrast to the other investigated cultivars, 'Candonga' contained only cya-3-glc, pel-3-glc, and pel-3-rut in large quantities, while cya-3-mal-glc and pel-3-mal-glc were only found in very small amounts in some locations and years (on average <0.4 and 0.7 mg kg$^{-1}$ FW). Each cultivar had an individual anthocyanin profile that will be discussed later (see Section 3.3). In general, the values for total and individual anthocyanin are in similar ranges as previously reported for these cultivars [11,12,14,21,30–32].

### 3.3. Anthocyanins are Affected by Location

To better characterize the influence of location and thus mainly latitude and climate, as well as yearly weather parameters on the anthocyanin profile of six strawberry cultivars, a cultivar-specific PCA was conducted. The score and loading plots comprised by the first two principal components (PC1 and PC2) explained ~70% of the total variance of the data set for each of the four cultivars ('Clery', 'Frida', 'Gariguette' and 'Sonata') grown in five locations (Figures 2 and 3), and of the two cultivars ('Candonga' and ''Florence') grown in four locations (Figure 4).

In addition, the PCA models also captured the effect of the two years and the two picking dates within each year. PC2 described 15.7–25.3% of the data variation and was mainly responsible for the separation of each cultivar by location, and thus by latitude and climatic factors, as well as local pre-harvest weather conditions on the anthocyanin synthesis and accumulation.

In general, the loading plots (Figures 2–4) of each cultivar indicates that pel-3-gluc are closely related to total anthocyanin, indicating a similar reaction on the different locations and their environmental conditions. Likewise, cya-3-glc and cya-3-mal-glc were grouped closely together. Thus, they were similar, but in an opposite way as pel-3-glc and total anthocyanins, influenced by locations and yearly weather variations. An exception was observed for 'Frida' where the total anthocyanin and pel-3-glc were clustered together with both cya-derivates, and hence, underlying the same location and environmental effects. Even though all pel-derivates have the same synthesis pathway, pel-3-rut and pel-3-mal-glc showed a cultivar and location-dependent distribution pattern. In the case of 'Clery', 'Gariguette' and 'Candonga', pel-3-rut was clustered more or less separately between both cya-derivates and total anthocyanin and pel-3-glc, while in 'Sonata', it was more related to the cya-derivates, in 'Frida', to cya-derivates, pel-3-glc, and total anthocyanin, while in 'Florence', pel-3-rut was linked to total anthocyanin and the other two pel-derivates. Pel-3-mal-glc was more or less related to pel-3-glc and total anthocyanin in the case of 'Clery', 'Gariguette' and 'Sonata'. In contrast, pel-3-mal-glc was clustered separately for 'Frida' and together with all other pel-derivates and total anthocyanins in the case of 'Florence'.

The score plots (Figures 2–4) indicated a similar clustering of location for each of the cultivars, showing a typical latitudinal division by the North-South axis. However,

modification also occurred by cultivar, yearly variations and weather conditions during the harvest period.

The high amount of cya-3-glc and cya-3-mal-glc in 'Clery' samples from Poland and Norway (Tables 3 and 6), and the enriched levels of total anthocyanins, pel-3-glc, and pel-3-mal-glc and in fruits grown in France (Tables 2, 4 and 7), mainly contributed to the clustering of the different locations (Figure 2). The pel-3-rut content was high in 'Clery' fruits from Germany (Table 5), however, this was less important for the separation by location. Interesting to note is the effect of yearly variations combined with harvest period on the synthesis and accumulation of total anthocyanins and their individual compounds at the different locations. For example, 17A samples from France are overlapping with those from Italy in 2018. Another example is the clear separation of the Polish 18B samples from their other ones.

The loading plot for 'Frida' (Figure 2) separated the locations more clearly than for 'Clery'. However, the effects of yearly variations and harvest period are less pronounced. The main difference between locations were due to the high abundance of pel-3-glc in samples from Norway in both years (Table 4), and total anthocyanin both in Norway and Poland (Table 2). In addition, clustering was due to low levels of cya-3-glc in fruits from Italy, especially in 2017 (Table 3), and low values of pel-3-rut (Table 5) in both Italy and France in 2018. Moreover, pel-3-mal-glc (Table 7) was found to be relatively less abundant in samples from Norway.

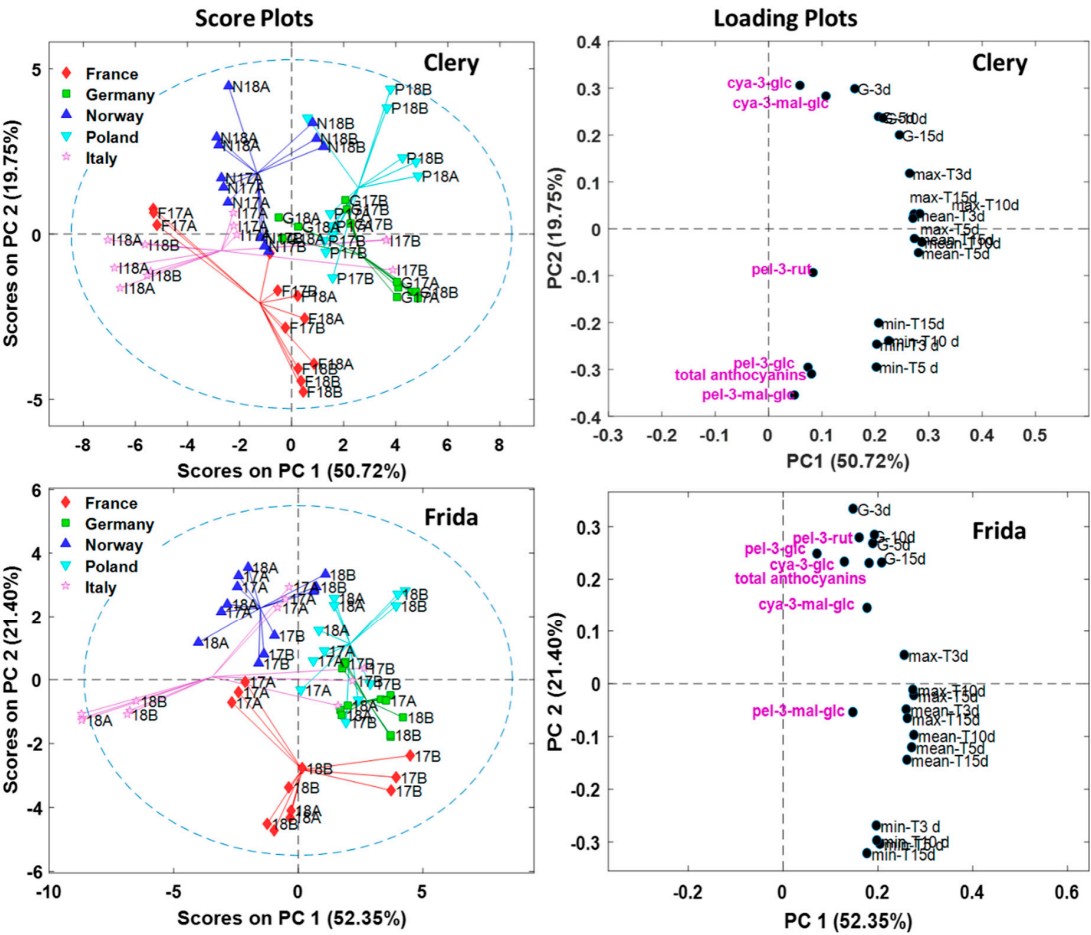

**Figure 2.** Score and loading plots of PCA for 'Clery', and 'Frida' grown at five locations (F: France; G: Germany; N: Norway; P: Poland and I: Italy) characterized by total and individual anthocyanin and climatic data. The number and capital letters in the different panel are referring to the year (2017 and 2018, labeled as 17 and 18) and A or B to the harvest date, being one week apart.

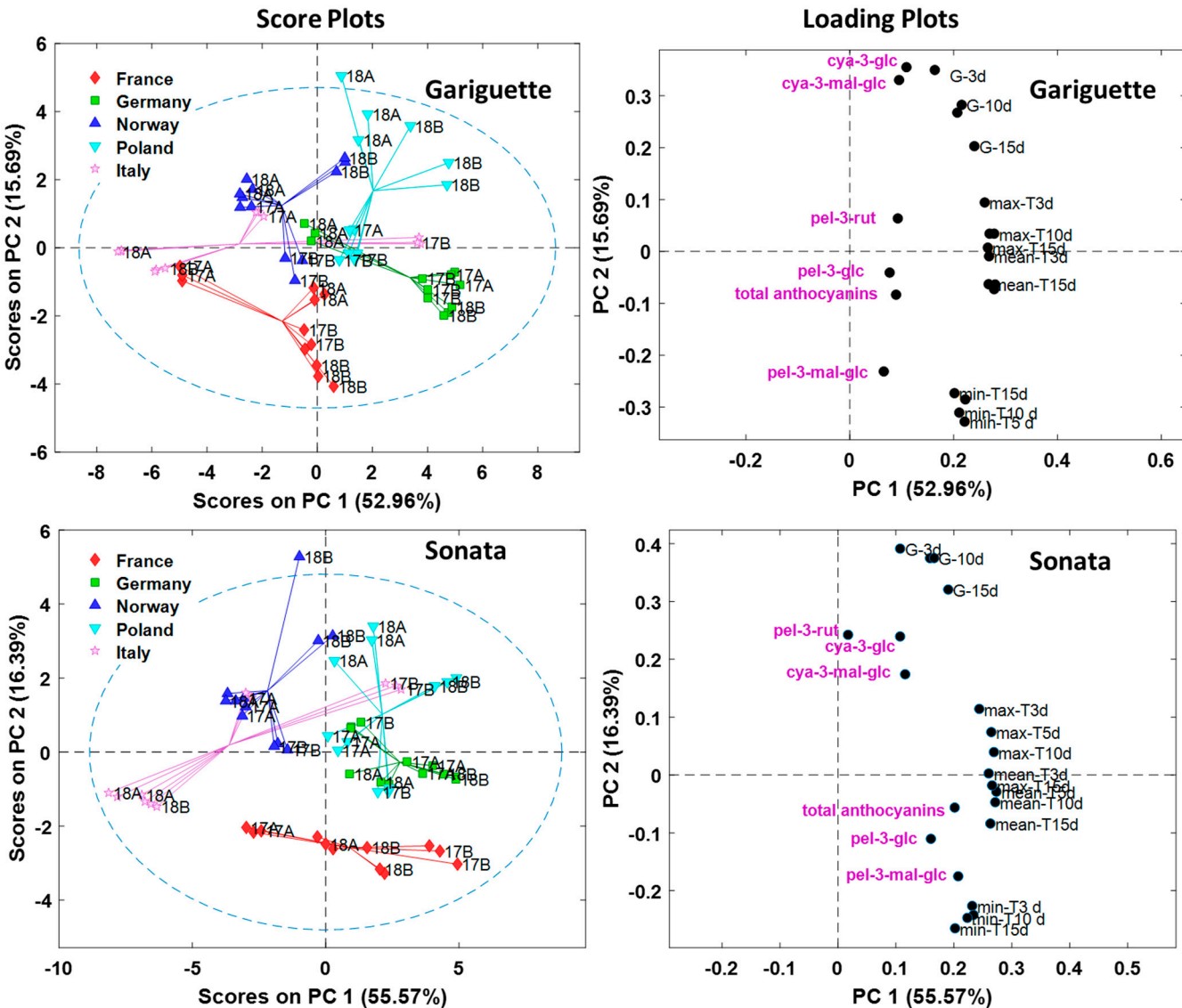

**Figure 3.** Score and loading plots of PCA for 'Gariguette', and 'Sonata' grown at five locations (F: France; G: Germany; N: Norway; P: Poland and I: Italy) characterized by total and individual anthocyanin and climatic data. The number and capital letters in the different panel are referring to the year (2017 and 2018, labeled as 17 and 18) and A or B to the harvest date, being one week apart.

Although the mean levels of total and individual anthocyanins in fruits of 'Gariguette' did not vary much between locations, as was the case for the other cultivars, the different locations were separated as well (see score plot in Figure 3). This was mainly due to the high level of cya-derivates in Poland in 2018 (Tables 3 and 6), and high amounts of pel-3-mal-glc in samples from France in both years, from Norway and Italy in 2017 and from Germany in 2018 (Table 7). Interestingly, total anthocyanin (Table 2) and its main component pel-3-glc (Table 4) did not contribute much to the separation of the locations because of contrasting year-on-year effects mainly in Norway, and even more pronounced, in Italy. In general, the anthocyanin profile of 'Gariguette' seemed to be less affected by the growing locations.

In the case of 'Sonata', total and individual anthocyanins did not vary much between the locations (Tables 2–7). Samples from France were clearly separated from the other locations (score plot in Figure 3), properly due to their relative low level of cya-derivates (Tables 3 and 6). Interestingly, Italian samples contained similar levels of these anthocyanins, but also less pel-3-glc (Table 4), and thus total anthocyanin (Table 2) in 2018, and this

distinguished these samples from those of France and partly from the other locations. Moreover, in 2018, Norwegian samples were enriched in pel-3-rut (Table 5), but contained, at the same time, low levels of pel-3-glc (Table 4), and total anthocyanin (Table 2) like samples from Italy. Both variables were of the factors responsible for the clustering of the Norwegian samples.

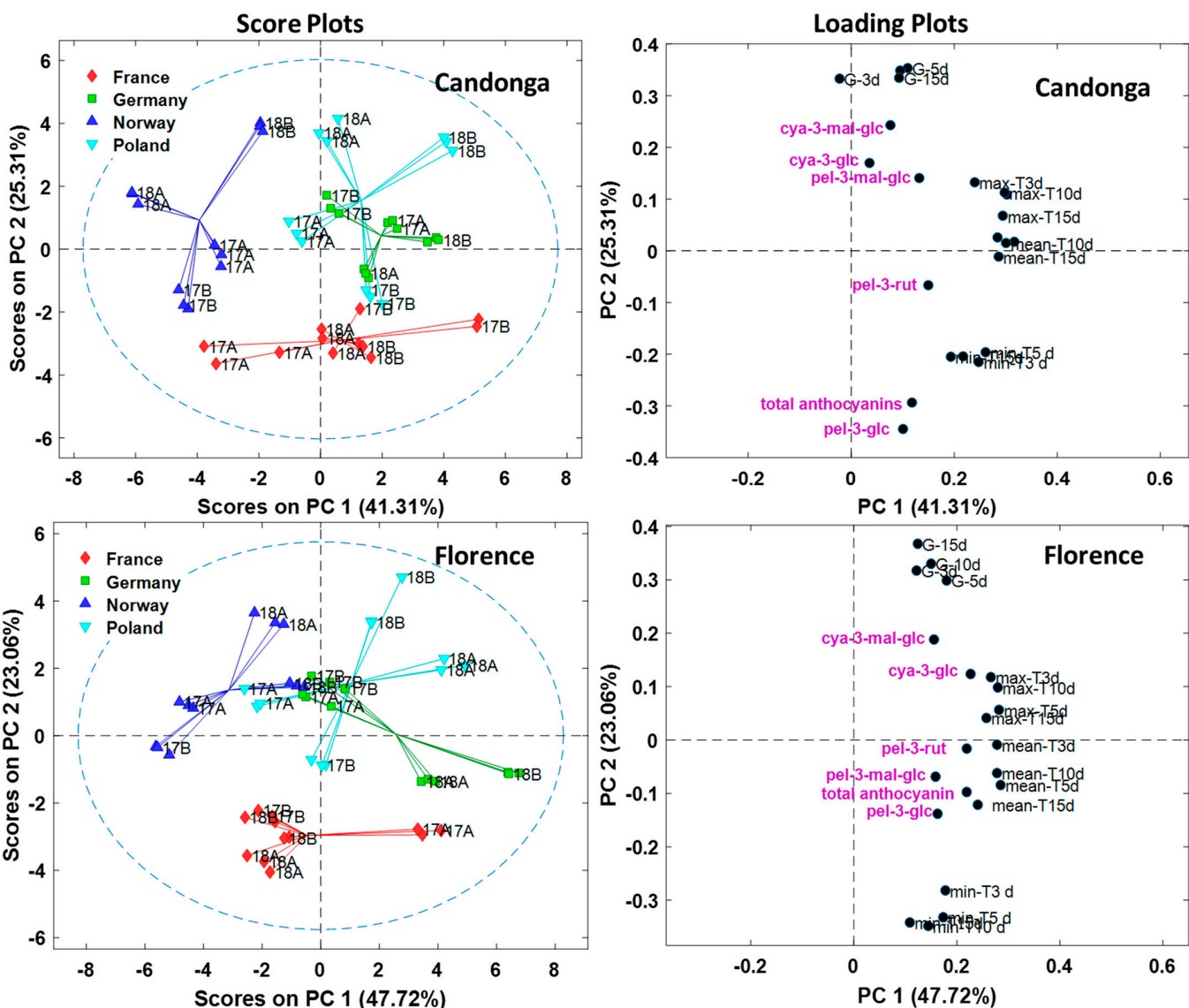

**Figure 4.** Score and loading plots of PCA for 'Candonga', and 'Florence' grown at four locations (F: France; G: Germany; N: Norway and P: Poland) characterized by total and individual anthocyanin and climatic data. The number and capital letters in the different panel are referring to the year (2017 and 2018, labeled as 17 and 18) and A or B to the harvest date, being one week apart.

Total anthocyanin and pel-3-glc were abundant in fruits of 'Candonga' grown in France but less enriched in those grown in Norway in 2018 (Tables 2 and 4). Thus, both compounds contribute to the separation of France from Norway, and both of these from the two other locations (Figure 4). Moreover, pel-3-rut (Table 5) was low in samples grown in Norway whereas samples grown in Poland were enriched in cya-3-glc (Table 3) and pel-3-mal-glc (Table 7) in 2018, and thus showing clear seasonal effects.

Fruits of 'Florence' grown in France were clearly separated from those grown in Norway (Figure 4). Moreover, PCA segregated both these locations from Germany

and Poland. In both seasons, the level of pel-3-glc and anthocyanins were high in the French samples, whereas seasonal effects only enriched the levels of total anthocyanins in Poland and Germany, of pel-3-glc in Germany and of pel-3-mal-glc in all locations in 2018 (Tables 2, 4 and 7). In addition, fruits grown in Poland and Germany were enriched in cya-derivates in 2018, too (Tables 3 and 6).

In general, the PCA separated samples of all cultivars from France from those from the other locations. One reason might be the low values of global radiation for France in both years (Figure 1). However, more reasonable is the fact, that here, as in Italy, strawberries were cultivated in an open-sided tunnel covered with a standard plastic film that is well known to be non-transparent for UV radiation. Among others, flavonoid syntheses in plants is strongly induced by light and UV-B wavelength (280–315 nm). They are effective scavengers of reactive oxygen species (ROS) and absorb selectively UV radiation [17]. One of these flavonoids are anthocyanins being synthesized in higher amounts by excess UV-light. Previous studies reported a retarded coloring of the ripening fruit resulting in a decreased level of total anthocyanin in strawberries grown under UV opaque film compared to UV transparent film [26]. Moreover, no effect of UV radiation on total anthocyanin were observed for strawberry, raspberry and blueberry when grown under films varying from UV blocking to highly transparent [24]. In a study by Josuttis et al. [23], it was shown that the anthocyanin cya-3-glc decreased in strawberries when grown under a UV- blocking plastic film. Cya-3-glc is a minor anthocyanin in strawberry cultivars but abundant, for example, in red *Lettuce sativa* types [24,33]. Additionally, enhanced levels of the derivate cya-3-galactoside were found in the skin of apple [34,35] when exposed to low night temperature and light including UV wavelength. For peach, a different genetic background-dependent cya-3-glc level was detected in two peach cultivars after postharvest treatments with UVA or UVB light [36]. Accompanying transcriptomic studies identified different cultivar-specific expressed genes related to anthocyanin synthesis. In the current study, a cultivar-dependent reaction to UV exclusion was observed in the way that only fruits of 'Clery' and 'Sonata' showed a decreased cya-3-glc content under the UV-blocking tunnel production in France and Italy compared to those from open-field production in Norway, Poland and Germany.

The fact that the Italian samples of each cultivar were not clustered like the French samples but located between France and the other locations, may be due to the lower temperature during the harvest period (Table 1) in Italy, because of latitude and altitude. Thus, temperature effects probably modified the UV-reducing tunnel effect in samples from Italy.

### 3.4. Impact of Temperature and Global Radiation on Cultivar-Specific Anthocyanin Profiles

As shown by the PC analysis, the synthesis and accumulation of total and individual anthocyanin were influenced by temperature ($T_{mean}$, $T_{max}$ and $T_{min}$) and global radiation and its interactions, being altogether affected by the latitude of the growing location. Environmental factors changing with latitude are mainly photoperiod, quantity and spectral composition of the solar radiation [17], as well as air temperature being indirectly dependent on solar radiation. However, weather conditions may vary between seasons at site and during the harvest period. Consequently, linear regression analyses were assessed to better explain the dependency of the cultivar-specific anthocyanin synthesis and accumulation on $T_{mean}$, $T_{max}$, $T_{min}$ and global radiation. Moreover, to evaluate the effective time span of these factors, they were summarized 3, 5, 10 and 15 days prior to harvest. Overall, the percentage of variations (Figure 5), explained by these environmental factors, were rather low with some exceptions (highest $R^2 = 0.62$), highlighting again, the genotype specific-based anthocyanin syntheses. For example 'Gariguette' and 'Clery', with the exception of cya-3-mal-glc, were not affected by the environmental factors tested. For 'Clery', this result supports previous studies showing also no or little environmental effects on anthocyanin accumulation in fruits from this cultivar, when grown at three locations from North to Central Europe [21], and at different altitude in Switzerland [31]. In contrast, 'Florence',

'Frida' and 'Sonata' reacted most sensitively, but differently, to the pre-harvest temperature and global radiation conditions. For these cultivars, in general, ($T_{max}$) had a higher impact on the syntheses of the less abundant anthocyanins cya-3-glc, pel-3-rut, cya-3-mal-glc, pel-3-mal-glc and on the total anthocyanin, than the minimum temperature. Thereby, the influence of $T_{max}$ often seemed to be stronger than the related $T_{mean}$ itself (for example, for the cya-derivates in 'Florence' and 'Frida'). Only the synthesis of pel-3-glc, the main anthocyanin of strawberry, was slightly more influenced by $T_{min}$ than by $T_{max}$. In the current study, global radiation was positive correlated with cyl-3-glc and pel-3-rut only in fruits of 'Frida'.

Noticeable is the contrasting behavior of 'Candonga' shown as the only cultivar with a negative relationship between global radiation and the main anthocyanin pel-3-glc ($R^2 = 0.38 - 0.44$), and thus also to the related total anthocyanin, and between cya-3-mal-glc and $T_{min}$. 'Candonga' was bred and selected for tunnel production in Spain, which is the common production system for that area. Therefore, it is assumed that protection against UV-radiation was not an important growing factor for this genotype. In a Spanish study performed in tunnels, no correlation was found for 'Candonga' between total anthocyanin and $T_{mean}$, $T_{max}$ and solar radiation and, in contrast to this study, a positive relationship to $T_{min}$ [14]. In the same study, however, other cultivars reacted also on $T_{mean}$ or $T_{max}$.

Noteworthy also is the positive relationship between global radiation and the content of cya-derivates in the Scandinavian cultivar 'Frida'. Moreover, 'Frida', together with 'Florence'—bred in the United Kingdom—were both sensitive to $T_{max}$ with high $R^2$ values for cya-derivates and pel-3-rut, and in the case of 'Frida', also of pel-3-mal-glc. Thus, both cultivars showed a high adaption to their breeding locations to benefit best from $T_{max}$ in areas with, in general, lower temperature, and in case of 'Frida', also from the latitudinal long photoperiod in Norway.

Numerous studies evaluated the effect of temperature on the anthocyanin content of strawberry. In general, the content of total anthocyanin was enhanced with increasing temperature when plants were grown under controlled conditions [9,19,20]). However, the major anthocyanin pel-3-glc was less affected than the minor abundant pel-3-mal-glc [19]. Moreover, increased night temperature induced higher anthocyanin levels [9]. A contrasting result was obtained in a study with Japanese cultivars where high growing temperature (30/15 °C day/night) decreased the anthocyanin content compared to the control (20/15 °C), due to differently expressed genes involved in the anthocyanin synthesis [37].

There are only few studies evaluating latitudinal or altitudinal effects on the anthocyanin content of strawberry. In general, although pre-harvest weather conditions modified the data, fruits grown at higher latitudes had less anthocyanins than the southern ones [13,22,38]. In addition, quantitative changes in the anthocyanin profiles were found [20], giving northern fruits a higher percentage of minor anthocyanins like cya-3-glc and cya-3-mal-glc. These latitudinal depending results confirm our findings. In a study comparing different altitudes (differences ~600 m) with increased temperature but lower radiation in the period 10 days before harvest at the higher altitude, no influence on total anthocyanin and pel-3-glc were found, but cultivar specifically increased levels of cya-3-gluc and minor pel-derivates [31]. In contrast, Guerrero-Chavez et al. [39] found a decrease of total anthocyanin, mainly pel-3-glc and an unnamed pel-derivate, with increasing altitude (~600 m) in fruits of 'Elsanta'. Furthermore, in a recent study, temperature, UV radiation and sunshine duration were found to affect bioactive compounds of strawberry stronger than locations differing nearly by 800 m altitude. However, anthocyanin was the compound class that showed significant differences between locations in one cultivar only [32].

In this experiment, 'Florence' and 'Frida' showed a high adaptation to their breeding place, and in case of 'Candonga', to the tunnel cultivation technique used during selection. High adaption ability is well known for wild species. For example, wild populations of different *Vaccinium* species, grown from South to North Scandinavia, showed significant variations in the anthocyanin profile and total anthocyanin content giving northern populations a higher anthocyanin content in their berries [40–42]. It was explained by the

long photoperiod at northern sites and its intense radiation of UV, visible and far-red wavelength [42], but also by its lower mean temperature or by the interaction of both. The adaption was under strong genetic control and remained when cloned plants of the Nordic populations were grown in South Scandinavia [42].

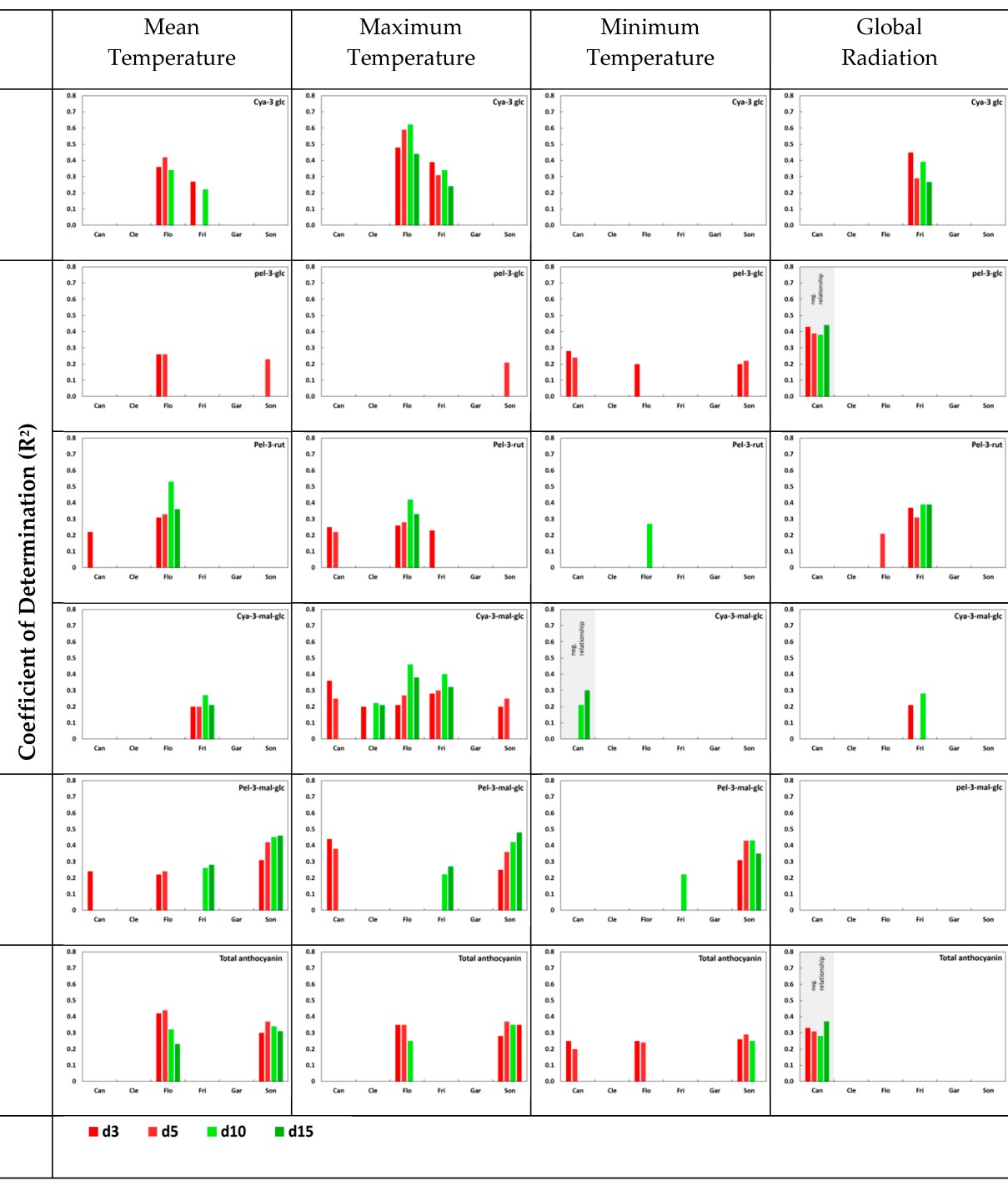

**Figure 5.** Coefficient of determinations ($R^2$) between cya-3-glc, pel-3-glc, pel-3-rut, cya-3-mal-glc, ple-3-mal-glc and total anthocyanin (shown top to bottom) and the environmental factors' mean temperature ($T_{mean}$), maximum temperature ($T_{max}$), minimum temperature ($T_{min}$) and global radiation accumulated for 3, 5, 10 and 15 days (3d, 5d, 10d and 15d) before sampling of the berries to be analyzed for each cultivar 'Candonga' (Can), 'Clery' (Cle), 'Florence' (Flo), 'Frida' (Fri), 'Gariguette' (Gar) and 'Sonata' (Son), shown left to right. Cultivars with no data (blank) indicate the observed correlation between different anthocyanin components, and mean, maximum, minimum temperatures and global radiation were statistically insignificant.

The pigmentation of strawberry occurs relatively rapid at the end of the fruit development. Metabolomic studies indicate the first appearance of anthocyanins with rapidly increasing amounts until ripeness around 5–10 days after the fruit's white stage [38,43–45]. These previous studies have focused on fruit developmental stages and total anthocyanins only [38,44], or cya-hexose, pel-hexose and pel-rutinose [45]. The current study considered different time intervals up to 15 days prior to harvest and took into account not only the total anthocyanin but all individual anthocyanins evaluated. As expected, pel-3-glc were cultivar-specific, and presented only the last five days before harvest. However, in our study, it was surprising to find environmental effects occurring at an earlier fruit developmental stage affecting the different anthocyanins. For example, in 'Candonga' fruits, global radiation inhibited its synthesis up to 15 days pre-harvest. Furthermore, 'Candonga' was the exclusive cultivar where pel-3-rut, cya-3-mal-glc and pel-3-glc were not found for $T_{max}$ at d10 and d15 before harvest whereas they were partly present in 'Florence', 'Frida' and 'Sonata'. The synthesis of the other individual anthocyanins was affected by temperature and less by global radiation when taking the whole period into account. It is assumed that at this early stage, photosynthesis was enhanced by favorable temperature and radiation conditions and, in that way, precursors of the anthocyanins like carbon skeletons were accumulated. When evaluating primary and secondary metabolites during strawberry development [45], a decrease of diverse sugars was found over time, while anthocyanins increased at the late fruit developmental stages.

Herein, the impact of location on anthocyanin profiles of certain cultivars was systematically studied. The obtained results showed that the present work could serve as a useful starting point towards evaluating the latitudinal effects on plant performance and internal fruit quality. However, some further specific studies are required to validate this. In addition, certain optimizations are still required for accounting the seasonal variations in the global radiation and temperatures. These issues will be addressed in detail in our near future research work.

## 4. Conclusions

Our study indicated that the anthocyanin content of strawberry cultivars are, beside the well-known genetic origin, partly affected at site by the local environmental factors, namely temperature, global radiation and cultivation technique. While 'Clery' and 'Gariguette' displayed a very high stability in their anthocyanin content regardless of the growing location, the other cultivars partly reacted on the local conditions of the growing sites. Thus, a high cultivar x environment interaction was observed for the evaluated cultivars. $T_{max}$, $T_{min}$ and global radiation, relatively less affected pel-3-glc. Cya-3-glc, cya-3-mal-glc, pel-3-rut, pel-3-mal-glc were found to be highly sensitive to $T_{max}$. In addition, global radiation strongly increased cya-3-mal-glc and pel-3-rut in case of 'Frida', while in case of 'Candonga', the abundance of pel-3-glc decreased with global radiation. The anthocyanin profiles of 'Gariguette' and 'Clery' were unaffected by environmental conditions.

The minor strawberry anthocyanins cya-3-glc, pel-3-rut, cya-3-mal-glc and pel-3-mal-glc seemed to be cultivar-specific more sensitive to such environmental variations than the abundant pel-3-glc. Thereby, the minor anthocyanins might be useful to breed cultivars which are able to accumulate anthocyanins even under sub-optimal conditions, for instance, like 'Frida' which produced high content of anthocyanins in Norway being sensitive to $T_{max}$ and global radiation. In contrast, cultivars with high stability in their anthocyanin content like 'Clery' and 'Gariguette' may be valuable parents as well in breeding programs focusing on the challenge of increased temperature due to climate change and on weather instability between and within harvest periods. Thus, a better understanding of the cultivar x environmental interactions will be necessary. However, the cultivar-specific relationship between fruit anthocyanin content and the evaluated environmental factors in our study were rather low, indicating that other factors not considered were involved.

**Supplementary Materials:** The following are available online at https://www.mdpi.com/2076-3417/11/3/1326/s1, Table S1: Sum of mean, maximum and minimum temperature as well as sum of global radiation, 3, 5, 10, and 15 days prior to sampling of fruits for analyses of 6 cultivars from North to South of Europe, Figure S1: PCA score plots for (a) the 6 cultivars (Clery (Cle), Candonga (Can), Frida (Fri), Florence (Flo), Gariguette (Gar) and Sonata (Son)) in 4 locations and (b) the 4 cultivars (Clery (Cle), Frida (Fri), Gariguette (Gar) and Sonata (Son)) in 5 locations clearly indicate that the location has major impact than the genetic variation. The '17' and '18' indicate years 2017 and 2018, respectively. The 'A' and 'B' indicate the two picking dates. In order to have better understanding, it is essential to analyze each cultivar separately.

**Author Contributions:** Conceptualization, E.K.; data curation, K.K.; investigation, E.K., F.W., P.C., A.M., D.M., A.P., G.S., K.C. and A.S.; writing—original draft, E.K., F.W. and K.K.; writing—review and editing, P.C., A.M., D.M., A.P., G.S., K.C. and A.S. All authors have read and agreed to the published version of the manuscript.

**Funding:** This research was funded by European Union's Horizon 2020 research and innovation programme, grant number 679303.

**Institutional Review Board Statement:** Not applicable.

**Informed Consent Statement:** Not applicable.

**Data Availability Statement:** Not applicable.

**Acknowledgments:** The authors would like to thank all farm workers and laboratory stuff for skillful field management, sample collection and analyses, respectively.

**Conflicts of Interest:** The authors declare no conflict of interest. The funders had no role in the design of the study; in the collection, analyses, or interpretation of data; in the writing of the manuscript, or in the decision to publish the results.

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
