# Peer review of "Influence of Post-Flowering Climate Conditions on Anthocyanin Profile of Strawberry Cultivars Grown from North to South Europe"

_applsci, doi:10.3390/app11031326_

Round 1
Reviewer 1 Report
This is an interesting work that has spanned a couple of years. The study can be improved and made clearer if authors can carry out corrections suggested below.
The abstract is too long. Reduce to the 200 words limit. It's 357 at the moment
Line 77: Authors stated that ....'In future, due to the climate changes, the northern regions of Europe will become more suitable for berry production.' This is supposition. Remove or provide evidence.
Line 102: Authors state 'strawberry genotypes used in this study were selected for their diverse genetic background and adaptability to different environments'... How did the authors know this? Any references?
Line 111: Also here......Where exactly? Norway, Poland, Germany, Italy or France?
Line 115: Authors stated To evaluate the impact of latitude-related factors on the anthocyanin synthesis and accumulation, daily temperature (Tmean, Tmax and Tmin) and global radiation were accumulated'.. Is this the standard or validated method for evaluating the impact of latitude? Why was no reference provided to support method chosen?
Line 129: Replace 'weight' with 'weighed'
Line 178-180: Revise English
Lines 229: Authors stated that these results are in consistency with the studies of Cocco et al. [12], Crescente et al. [13] and Josuttis et al. [20]. What exactly was consistent? This approach was also used in line 240. Authors need to be more specific.
Line 332: Replace 'analyses' with 'analysis'
Most PCA plots in Fig 2 and 3 are unreadable. Put fewer plots per page and make panels bigger for clarity.
Authors did not use the manuscript template as directed. It made following the R & D difficult.
The link provided for reference 1 does not lead to the information provided in text hence could not be verified. Authors should ensure that they are on the page that has the information before providing the online link as per the guide.
References have not been prepared as per the journal guide. Reference 1 need to be revised using the journal's guide. For the rest of the references, authors have not made sure journal titles and volumes are in italics. The guide is explicit on how these should be written, so these should be corrected accordingly.
Authors stated in abstract 'The anthocyanin profiles of the different cultivars varied significantly and were mainly genetically inherited' but the title of the study is ..'Influence of post-flowering climate conditions on anthocyanin profile'... Also, the first line of the conclusion states 'Our study confirms that the anthocyanin content of strawberry cultivars is strongly dependent on their genetic properties'...It appears the title should be .'Influence of genetic variation.......
The other argument that the result shown in this study is due to climate conditions is weak. Use genetic variation instead as the main finding and reflect this in the title
Another issue is Thus, a high gene x environment interaction was determined..
Authors have not demonstrated this. For a start which cultivar had the overall best result and what was the performance across locations? This information is missing in the abstract and conclusion.
Reviewer 2 Report
This study is about changes in strawberry anthocyanin content in various growing strawberry regions. It would be better if you do not put a point on the cultivation area and describe the difference in the environment of the cultivation area as the point. Please refer to the attached file and modify it.

Author Response
Please see the attchment.

Round 2
Reviewer 1 Report
Authors response: Now line 102: The authors performed strawberry trials for years and tested cultivars from the different origin of Europe and different breeding programs. Thus, even not always references are available, they know the selected cultivars very well.
If the authors have performed strawberry trials for years, there should be a report somewhere, which can be used to verify the information supplied. Also If authors have chosen the cultivar based on personal experience, then it should be included in the text so that any investigator that wants to use the same cultivar will know that it is only the authors' opinion.
Response – now line 117: To the best of our knowledge, there is no explicit method to evaluate latitudinal effects on plant performance and internal fruit quality.
How will the conclusions be evaluated if the method used has not been scientifically validated?
Response: We checked the online link 10 January and http://www.fao.org/faostat/en/#data/QC is the right. It is an interactive page and people have to chose the crop (strawberry),
The norm is for authors to provide references used. Weblinks must lead to the information provided. Authors should provide a link to the page which has the information.
Response: We will not change the title because the genetic influence on anthocyanins is well known
The title needs to be amended to reflect the work carried out. Figure 5 sets out the environmental factors recorded for the study and the variation is evident. Are the cultivars top to bottom? Two graphs are blank. An explanation should be given in the legend to explain the blank panels.
Authors stated in lines 252-253 - Since temperature and global radiation may vary within the harvest season .................... How did the authors control seasonal variations?
Response Lines 213-216: The PCA analysis carried out on all the cultivars together clearly indicated (not shown) that the anthocyanin profiles of the strawberries are mainly influenced by the environmental factors (latitude, longitude, temperature etc.)
The data that is supposed to back up the title and the main conclusion is not shown. Not even as a supplement? Also, the authors reported that auto-scaling was carried out before PCA. One of the main limitations of PCA is scaling. How did authors mitigate this?
In my opinion, it is difficult to see the influence of climatic conditions in the results presented.
Round 3
Reviewer 1 Report
Most concerns have now been addressed. Authors should just add the limitations of the methods used.
